# Changes in the VOC of Fruits at Different Refrigeration Stages of ‘Ruixue’ and the Participation of Carboxylesterase *MdCXE20* in the Catabolism of Volatile Esters

**DOI:** 10.3390/foods12101977

**Published:** 2023-05-12

**Authors:** Dongmei Li, Jianhua Guo, Hai Ma, Linna Pei, Xiaojie Liu, Hui Wang, Rongxin Chen, Zhengyang Zhao, Hua Gao

**Affiliations:** College of Horticulture, Northwest A & F University, Xianyang 712100, China; lidongmei@nwafu.edu.cn (D.L.); guojianhuahua@nwafu.edu.cn (J.G.); mahai@nwafu.edu.cn (H.M.); peilinna@nwafu.edu.cn (L.P.); echoliu@nwafu.edu.cn (X.L.); zhaozy@nwsuaf.edu.cn (Z.Z.)

**Keywords:** ‘RuiXue’, apple, ester, *MdCXE20*

## Abstract

Aroma is a crucial quality attribute of apple fruit, which significantly impacts its commercial value and consumer choice. Despite its importance the volatile aroma substances produced by the new variety ‘Ruixue’ after harvest remain unclear. In this study, we utilized headspace solid phase microextraction-gas chromatography-mass spectrometry (HS-SPME-GC-MS) to investigate the changes in volatile substances, fruit hardness, crispness, and related aroma synthase activity of commercially mature ‘Ruixue’ apples during cold storage. Our findings revealed a gradual decline in fruit firmness and brittleness of ‘Ruixue’ apples during cold storage, with hexyl acetate, hexyl caproate, and hexyl thiocyanate being the main hexyl esters detected. To gain a better understanding of the metabolic pathway of esters, we identified 42 MdCXE gene members that are associated with ester degradation. Through RT-qPCR analysis, we discovered that carboxylesterase *MdCXE20* exhibited higher expression levels compared to other MdCXE genes during cold storage. To confirm the role of *MdCXE20*, we conducted a transient injection of apple fruits and observed that overexpression of *MdCXE20* led to the degradation of esters such as hexyl hexanoate, butyl hexanoate, butyl 2-methylbutyrate, hexyl butyrate, and hexyl 2-methylbutyrate. The results of the study showed that the virus-induced gene silencing of *MdCXE20* found the opposite results. Additionally, the esters of OE-*MdCXE20* callus showed a lower content of ester VOC than the control callus, according to the homologous stable transformation of ‘Wanglin’ callus. Overall, these findings suggest that the *MdCXE20* gene plays a crucial role in the decrease of esters in ‘Ruixue’ apples, which ultimately affects their flavor.

## 1. Introduction

Apple *(Malus* × *domseica* Borkh.) is one of the four major fruits in the world. It is widely planted and loved by people because of its important edible value, economic value, social value and ecological benefits [1]. With the rapid development of economy, consumers are not only satisfied with external qualities such as appearance, color and mechanical damage, but also pay more attention to the internal quality of fruit nutrition and flavor. It is widely recognized that flavor is determined by a complex mixture of sugars, organic acids and volatile compounds (VOC). Among these, VOCs play a key role in determining fruit quality and consumer preference [2,3]. Presently, there are more than 2000 volatile aroma substances in plants, and more than 350 volatile aroma substances have been identified in mature apple fruits [3,4], including esters, alcohols, ketones, aldehydes and terpenes. The content of these volatiles is very low (ng/g), which is 1/1000 of the level of sugar acid (mg/g). Therefore, small changes in their components will greatly impact the flavor quality of apples. Among them, volatile esters are represent one of the important characteristics of apple fruit flavor [5,6]. For example, among the volatile compounds, methyl butyrate, ethyl butyrate, butyl butyrate, methyl caproate, ethyl caproate, butyl acetate, and hexyl acetate play an important role [7]. The synthesis of volatile substances in apple fruit is a dynamic process. During storage, the content and diversity of volatile substances in apples increased first and then decreased [8]. In order to prolong the fruit quality and preservation period and prevent decay after harvest, low temperature storage is usually used to inhibit the respiration, ethylene synthesis and pectin dissolution of apple fruit. However, it is found that the volatile substances of apple after cold storage have a certain impact, which seriously affects the commodity quality of apple [8]. Therefore, studying the aroma changes of chilled apples has important theoretical and practical significance for improving apple quality.

Esters are dominant in the aromatic substances of mature fruits such as apples, bananas, strawberries, peaches, and pears. As the fruit matures, esters increase significantly [9]. The specific components of ester aroma vary depending on the type of fruit. Apples are mainly composed of two types of esters, namely branched chain esters, like 2-methylhexyl butyrate and 2-methylbutyl acetate, and linear esters such as hexyl butyrate and hexyl acetate [10,11]; Ripe peach flavors are dominated by hexyl acetate, with linear esters such as vinyl ester E-2, being the most important [12]. Strawberry fruit is mainly composed of linear esters such as ethyl acetate and butyl acetate [13]. Ethyl acetate and butyl acetate are the main aromatic substances in Nanguo pears. Ester volatile substances have unique aroma, such as: butyl hexanoate has apple flavor, amyl acetate has banana, apple flavors; butyl butyrate has a fresh sweet fruit flavor [14].

Carboxylesterase (CXE, EC 3.1.1.1) is a metabolic enzyme found in a variety of organisms. It has the ability to catalyze the hydrolysis of carboxylic acid esters into corresponding alcohols and carboxylic acids. The function of this enzyme has been extensively studied in mammals, insects, and microorganisms [15,16]. Relatively speaking, there are relatively few studies on the CXE function of plants. The majority of these proteins are categorized under the α/β hydrolase folding superfamily. This particular superfamily shares a common core structure consisting of eight β chains that are interspersed with α-helixes and loops [17]. CXEs, in particular, have a catalytic triad that consists of serine (which is contained in the conserved common sequence GXSXG), an acidic amino acid, and histidine. This triad constitutes the active site of CXEs [17,18,19]. CXE enzymes exhibit a diverse range of substrate catalytic activity that is closely linked to various biochemical and physiological processes. These processes include the breakdown of exogenous substances and the conversion of volatiles, which play a crucial role in determining the aroma of mature fruits. Additionally, CXE enzymes are actively involved in plant secondary metabolism and plant defense responses [20,21]. Several studies have identified 20 CXE genes in the genome of *Arabidopsis thaliana*, which are found in various tissues [19]. Among these genes, *AtCXE8* overexpression in *Arabidopsis thaliana* has been shown to enhance plant resistance to gray mold invasion, while knockout of this gene renders plants susceptible to gray mold [22]. Additionally, tobacco *hsr203J* is involved in the detoxification of pathogen-derived compounds [23]. In a study conducted by [24], overexpression of *PepEST* in transgenic pepper fruits resulted in increased resistance to anthracnose. Fruit aroma and flavor are largely dependent on volatile esters, such as butyl acetate and hexyl acetate. *MdCXE1* in apple fruit was found to indirectly impact the hydrolysis of these esters, which are key components of mature fruit aroma [25]. Similarly, *SlCXE1* was identified as a regulator of tomato volatile acetate content [26]; functional analysis of 33 CXEs in peach fruit showed that 13 were expressed in fruit. The expression of six members (*PpCXE1*, *PpCXE2*, *PpCXE3*, *PpCXE6*, *PpCXE27* and *PpCXE32*) was related to ripening, and the transcription products of *PpCXE1*, *PpCXE2* and *PpCXE3* were the most abundant in mature fruits [27].

‘Ruixue’ is a significant apple variety in Shaanxi Province, and its quality is measured by its aroma. The aroma plays a crucial role in determining its economic prospects in the market. However, there is a lack of research on the aroma volatile substances of the new variety ‘Ruixue’ apple during cold storage. This study investigates the contribution of ester aromatic substances, including butyl acetate, hexyl acetate, and 2-methylbutyl acetate, to the aromatic quality of apple fruits. The hydrolysis of esters is facilitated by the carboxylesterase CXEs gene, which is known to play a crucial role in this process. However, there is a lack of experimental evidence regarding the function of this gene. To address this gap, the experimental material used in this study is ‘Ruixue’ apple fruit. This study investigated the changes in fruit hardness, brittleness, aroma key synthase activity, and volatile aroma substances in apples during cold storage. Additionally, the study screened the key gene *MdCXE20*, which is related to the hydrolysis of esters in apple fruits. This study provides a preliminary analysis of the changes in aroma and degradation of esters during cold storage. These findings serve as a basis for promoting a better understanding of these changes and their impact on molecular breeding in the future.

## 2. Materials and Methods

### 2.1. Plant Materials

The apple variety ‘Ruixue’ was collected in October 2020 at the Baishui Apple Experimental Station of Northwest A&F University (35°21′ N, 109°55′ E). A total of 150 fruits with the uniform size, uniform coloring, consistent maturity, no pests and diseases, and no mechanical damage were selected and bagged. After harvesting, they were placed in a room temperature (25 °C) environment for 12–24 h to dissipate the field heat. The next day, the plastic turnover box with single-layer paper and net belt filled with surrounding bedding newspapers (lined with fresh-keeping bags) was immediately transported back to the laboratory for processing. The fruits were stored at 1 ± 0.5 °C with relative humidity of 80–90% for 5 months, and sampled every other month for a total of 5 times. Each treatment had three biological replicates, and each biological replicate consisted of 10 fruits. After sampling, it was immediately frozen in liquid nitrogen and stored in a refrigerator at −80 °C for later use.

### 2.2. Determination of Fruit Physiological Characteristics

#### 2.2.1. Firmness Measurement by Texture Analyzer

The texture analyzer (FTC, Washington, DC, USA) was utilized to measure the firmness and brittleness of the fruit. The measurement parameters included a 10 mm depth, a P/2 probe of 2 mm, and a measurement speed of 2 mm/s. From each treatment fruit, five fruits were randomly selected, and the skin was measured five times on the equatorial line of the central yin and yang surfaces of each fruit. The maximum peak force value measured each time was firmness, and finally the average value was taken as kg/cm^2^. The ratio of the first peak force value to the movement distance was fruit brittleness, and the unit was Kg.sec.

#### 2.2.2. Determination of Lipoxygenase (LOX)

The method of Chen [28] is improved. Extraction of LOX crude enzyme solution: 5.0 g of liquid nitrogen fully ground fruit sample tissue was added to 10 mL 100 mM phosphate buffer solution (pH 7.5, containing 2 mM DTT, 0.1% (*v*/*v*) Triton X-100, 1% (*w*/*v*) PVPP) under ice bath conditions, and fully shocked and mixed. The mixed homogenate was centrifuged at 25,000× *g* for 15 min at 4 °C after passing through 4 layers of gauze, and the supernatant was taken as crude enzyme solution.

The reaction substrate used in the experiment was 10 mM sodium linoleate. Configuration method: 70 mg of sodium linoleate, 70 μL Triton X-100, and 4 mL of anaerobic water were mixed (to avoid bubbles), and then titrated with 0.5 mol/L sodium hydroxide to clarify the solutio. The solution was diluted to 25 mL, packed into a 1.5 mL centrifuge tube, and stored at −20 °C.

Determination of lipoxygenase activity. In the 3 mL reaction system containing 25 μL of sodium linoleate mother liquor, 0.1 mM (pH 6.0) citric acid-phosphate buffer 2.775 mL, reaction temperature 30 °C (keep 10 min), 0.2 mL of crude enzyme solution was added and mixed. After 15 s of reaction, the lipoxygenase activity was measured at 234 nm. The data of at least 3 points were obtained every 30 s, and the changes of OD value within 1 min were recorded and repeated three times.

#### 2.2.3. Determination of Hydroperoxide Lyase (HPL)

The method for extracting HPL crude enzyme solution was improved following Zhang [29]. 3.0 g of frozen fruit tissue was ground into powder in a mortar using liquid nitrogen. Then, 6 mL of pre-cooled extract at 4 °C containing 150 mM HEPES-KOH buffer (pH 8.0), 250 mM sorbitol, 10 mM EDTA, 10 mM MgCl_2_, 1% *v*/*v* glycerol, and 4% PVPP was added. The homogenate was fully mixed and extracted, and then centrifuged at 25,000× *g* (4 °C) for 30 min. The supernatant was collected as the crude enzyme solution.

The reaction substrate was sodium hydroperoxylinoleate. The configuration method: 10 mL distilled water, 200 μL 10 mM sodium linoleate, 400 μL LOX enzyme solution (1 mg/10 mL boric acid buffer, pH 9.0), with substrate at 30 °C in a water bath 2 h.

The activity of HPL was determined through a reaction system of 3.5 mL. The system contained 2 mL of analytical buffer consisting of 150 mM HEPES-KOH pH 8.0, 250 mM sorbitol, 10 mM EDTA, and 10 mM MgCl_2_. Additionally, the system contained 0.75 mL of substrate solution, 0.15 mL of 1.6 mM NADH, 0.1 mL of ADH enzyme solution (containing 1.5 mg/3 mL boric acid buffer, pH 8.6), and 0.5 mL of crude enzyme solution. The HPL activity was measured at 344 nm and 30 °C. The reaction was initiated 15 s after the addition of enzyme solution, and the change in OD value within 1 min was recorded and repeated three times.

#### 2.2.4. The Activities of Keto Acid Decarboxylase (PDC) and Aldehyde Dehydrogenase (ADH) Were Determined

The method of Ke [30] is improved. Extraction of crude enzyme solution: 5 mL 100 mM MES buffer solution (containing 1% (*w*/*v*) PVPP, 2 mM DTT, pH 6.5) was added to the tissue of 3.0 g fruit samples fully ground in liquid nitrogen under ice bath conditions, and fully shaken and mixed. The mixed homogenate underwent centrifugation at 4 °C and 25,000× *g* for 15 min after passing through 4 layers of gauze. The resulting supernatant was collected as the crude enzyme solution.

Determination of PDC enzyme activity: 0.45 mL 100 mM pH 6.5 MES buffer, 0.1 mL 5 mM TPP, 0.1 mL 50 mM MgCl_2_, 0.05 mL 1.6 mM NADH, 0.1 mL 50 mM pyruvate, 0.1 mL enzyme extract.

Determination of ADH activity: 0.8 mL 100 mM pH 6.5 MES buffer, 0.05 mL 1.6 mM NADH, 0.1 mL enzyme extract, 0.05 mL 80 mM acetaldehyde.

The absorbance change was measured at 340 nm for 2 min, starting 15 s after the enzyme solution was added, at a reaction temperature of 25 °C. The OD value change was recorded for each sample, which was repeated three times. The reaction solution without enzyme extract was used as a blank.

### 2.3. VOC Extraction and GC-MS Analysis

The determination was performed using Yang’s modified SPME-GC-MS method [31]. To prepare the sample, fresh fruits were ground into powder using liquid nitrogen. 4.0 g of this powder was weighed and added to a 50 mL screw-cap vial containing 10 mL of saturated NaCl and 10 μL of 0.4 mg/mL 3-nonanone, which was used as an internal standard. The headspace vials were equilibrated for 10 min at 55 °C on a metal heating and stirring platform. A divinylbenzene/carboxy/polydimethylsiloxane (DVB/CAR/PDMS) coated SPME fiber with a thickness of 50/30 µm (Supelco, Bellefonte, PA, USA) was inserted into the headspace for 30 min under continuous heating and stirring (200 rpm) to adsorb VOC. The fiber was then inserted into the heating inlet of the chromatograph and desorbed at 250 °C for 2.5 min. VOCs were analyzed using a Thermo Trace GCU ultra gas chromatograph (Thermo Fisher Scientific, Waltham, MA, USA) equipped with an HP-INNO Wax 60 m × 0.25 mm × 0.25 mm capillary column.

GC conditions: Helium was used as the carrier gas and circulated in splitless mode at a flow rate of 1.0 mL/min. The initial oven temperature was 40 °C, held for 3 min, then ramped to 150 °C at a rate of 5 °C/min, then ramped to 220 °C at a rate of 10 °C/min and held for 5 min.

MS conditions: mass spectrometry ion source, transfer line temperature 240 °C. Mass spectrometry was performed in electron ionization mode at 70 eV with a scan range of 35–450 *m*/*z*.

The volatile aroma substances were analyzed by GC-MS. The mass spectrometry data were automatically retrieved and matched with the NIST 2014 standard spectrum. The data with a matching degree of more than 60% and a matching value of more than 800 were retained to determine the chemical composition.

Quantitative analysis: The relative content of volatile substances [32] was obtained by peak area normalization method. The calculation formula was as follows: the content of each component (μg/kg) = (component peak area/internal standard peak area) × internal standard concentration (g/L) × 10 μL/sample mass (g).

### 2.4. RNA Extraction, Gene Expression Analysis by RT-qPCR

According to previous studies [33], the modified CTAB method was used to isolate and purify RNA from ‘Ruixue’ fruits at different storage periods under different treatments. After RNA extraction, cDNA was synthesized using the EasyScript One-Step gDNA Removal and cDNA Synthesis Kit according to the manufacturer’s instructions. The expression level of the candidate gene carboxylesterase, which is involved in the degradation of aroma esters, was analyzed using fluorescent quantitative real-time PCR (RT-qPCR) with cDNA as the template. Gene-specific primers were designed using Origin 2018 64 bit and NCBI. *MdActin* was selected as the reference gene for constitutive expression. All primer sequences are listed in Table 1.

### 2.5. Bioinformatic Sequence Analysis

The members of the MdCXE gene family all contain a conserved core structure, which is composed of eight β chains interspersed with α-helices and rings [17], and contains a catalytic triad composed of serine (contained in the conserved common sequence GXSXG), acidic amino acids and histidine, constituting its active site. Using this sequence, we searched the GDR apple genome database (https://www.rosaceae.org/, accessed on 5 March 2022) using BLASTP, and used *Arabidopsis* CXEs with 10^−5^ as the cut-off value to screen apple MdCXE gene family members. A total of 42 members of the MdCXE family were retrieved, and CDD belonged to Abhydrolase-3, Pfam07859. The GXSXG conserved sequence of the MdCXE gene family was analyzed by MEME online software (http://meme-suite.org/tools/meme, accessed on 5 March 2022). In this study, we employed several bioinformatics tools to analyze amino acid sequences, predict domain and function using MEME 6.0 and InterPro, respectively. To determine the physicochemical properties of some genes, we used the *p*I/Mw tool in ExPASy Compute. We also utilized Wolfpsort to predict subcellular location and calculated *p*I and Mw using the ExPASy Compute tool. Additional details can be found in Appendix A. The phylogenetic tree was constructed using MEGA 6.0. The maximum likelihood tree was tested and analyzed 1000 times for accuracy.

### 2.6. Transient Injection of the MdCXE20 in Apple Fruit

The apple fruit was infected by injection, method reference by Wu [34]. The full-length CDS of *MdCXE20* gene was amplified by PCR using the cDNA of ‘Ruixue’ fruit as a template and the primers in Table 2. The BamHI and KpnI restriction sites were selected and the pCAMBIA2300-OE-*MdCXE20* overexpression vector was constructed by homologous recombination. The 400 bp fragment of *MdCXE20* was cloned into TRV2 as a virus-induced silencing vector, and KpnI and EcoRI were selected as restriction sites. The TRV1 + TRV2-*MdCXE20* silencing vector was constructed by homologous recombination. The recombinant vector confirmed by sequencing was transformed into EHA105 Agrobacterium competent. Positive single colonies were selected and activated by YEP liquid culture containing kanamycin sulfate (50 μg/mL) and rifampicin (50 μg/mL). Based on 28 °C culture to OD_600_ = 1.2~1.5, the bacteria were collected by centrifugation, and the equal volume of sterile MS culture medium (10 mM MgCl_2_, 10 mM MES, 100 mM acetosyringone, pH 5.8) was added to re-suspend the bacteria. The empty pCAMBIA2300 and TRV1 + TRV2 vectors were used as controls, and the dark treatment was performed for 3 h. Choose the fruit with the same size and no mechanical damage, wipe the fruit surface with a wet cloth and wait for use. The apple fruit was divided into four parts by injecting the treated bacterial solution into the central axis epidermis of the fruit with a 1 mL syringe. Half of the fruits were injected with Agrobacterium containing pCAMBIA2300-OE-*MdCXE20*, and half were used for Agrobacterium infection containing pCAMBIA2300-OE, the silencing vector was the same as above. After labeling, it was placed in dark for 7 days, sampled, frozen in liquid nitrogen, and stored at −80 °C for VOC and gene analysis (15 fruits in each group).

### 2.7. Overexpression of ‘Wanglin’ Callus Transgenic

The Agrobacterium-mediated transformation method of ‘Wanglin’ callus is referenced [35]. The full-length CDS of *MdCXE20* gene was amplified by PCR using the cDNA of ‘Ruixue’ apple fruit as a template, combined with the primers in Table 2, and the CDS of *MdCXE20* gene were constructed by homologous recombination. The recombinant vector was introduced into Agrobacterium EHA105 through electroporation while the empty vector was utilized as a control. Positive single colonies were then chosen and grown in YEP liquid culture supplemented with kanamycin sulfate (50 μg/mL) and rifampicin (50 μg/mL). Based on the culture at 28 °C to OD_6oo_ = 0.6~0.7, the bacteria were collected by centrifugation, and the equal volume of sterile sterilized water containing 50 mg/L acetosyringone was added to re-suspend the bacteria, and placed in darkness for 3 h.

The callus of ‘Wanglin’, which grew for 20 days under sterile conditions, was infected with Agrobacterium containing pCAMBIA2300-OE-*MdCXE20* vector for 25 min. During the period of continuous shaking, the excess bacterial liquid was sucked with filter paper and placed on the subculture solid medium. After 3 days of dark culture, it was transferred to the screening medium containing 200 mg/L carbenicillin and 50 mg/L kanamycin sulfate, and cultured at 25 °C, 16 h/8 h in darkness. Observe their growth every 1 week until there is a small callus growth. Subculture medium components: 4.43 g/L MS, 7 g/L agar, 30 g/L sucrose, 0.5 mg/L 2,4-D, 1 mg/L 6-BA, pH 6.0. The differentiated callus was transferred to a new screening medium and subcultured every 2 weeks. DNA was extracted from the T8 generation of six lines for positive plant detection, and three lines were selected. The identified T8 transgenic ‘Wanglin’ callus was frozen in liquid nitrogen and stored at −80 °C for ester content and gene expression analysis. Three plants were selected from each line as three biological replicates, each containing five calluses. The callus *actin* gene was used as an internal reference gene to analyze the gene expression of *MdCXE20*.

### 2.8. Subcellular Localization Analysis

The analysis method is slightly modified based on [36]. The complete CDS sequence of *MdCXE20* was fused into the green fluorescent protein GFP to obtain the 35S-GFP-*MdCXE20* recombinant plasmid. The primer information is shown in Table 2. According to the method of described in Section 2.6, the fusion protein 35S-GFP-*MdCXE20* was transiently overexpressed in a *Nicotiana benthamiana* plant with MCherry nuclear localization marker by Agrobacterium infection. After 3 days of dark culture after injection, the laser confocal fluorescence microscope was used to observe and photograph.

### 2.9. Statistical Analysis and Data Processing

Excel 2010 software was used for data analysis, and IBM SPSS Statistics 26.0 software was used for one-way analysis of variance and significance analysis.

## 3. Results

### 3.1. Changes of Fruit Firmness and Brittleness of ‘Ruixue’ at Different Cold Storage Stages

Firmness is one of the important factors to evaluate apple quality [37]. As shown in Figure 1A, the firmness of apple fruit showed a decreasing trend with the increase of frozen days. The decrease of firmness of refrigerated fruit mainly occurred in the first 4 months of storage period, and the decrease was larger. The firmness of fruit decreased from 10.25 kg/cm^2^ to 7.67 kg/cm^2^ at 150 days of cold storage, which was 25.17% lower than that of harvest period. As shown in Figure 1B, the fruit brittleness showed a slow downward trend with the prolongation of storage time, indicating that refrigeration was beneficial to the fruit. It shows that cold storage can maintain fruit firmness quality attributes well.

### 3.2. Changes of Aroma Synthase Enzyme Activity in ‘Ruixue’ at Different Cold Storage Stages

C6 substances, such as 2-hexenal, are synthesized through the LOX-HPL pathway to produce hexyl esters. The content of 2-hexenal was the highest at 60 days of storage, the content of esters reached the maximum peak at 60 days of storage, and the ester substance 2-methyl hexyl butyrate also reached the maximum release amount, but the LOX and HPL activity reached the peak at 90 days (Figure 2A,B). It is speculated that this may be related to the content of linolenic acid and linoleic acid. It can be seen that the activity of PDC and ADH showed an overall upward trend in each storage stage during the whole storage period, and the rate of increase of enzyme activity was the fastest between 90 d and 120 d. The difference of PDC activity between 150 d and 1–30 d was the most obvious, and the difference of ADH activity between 120 d and harvest was the most obvious. (Figure 2C,D, *p* < 0.05).

### 3.3. VOC in Apple Fruits during Different Cold Storage Periods

In ‘Ruixue’ apple, the concentration of total aroma substances was a dynamic process during storage, which increased from 4839.45 μg/kg at harvest to 18,733.17 μg/kg at 60 days of cold storage (Figure 3A), and reached the maximum peak. A total of 51 volatile substances were detected during cold storage, and 36, 38, 48, 42, 43 and 40 aroma substances were detected respectively (Appendix A). With the prolongation of storage time, the types of aroma substances increased first and then decreased, among which the changes of esters were the most obvious. The types of esters during cold storage period were 19, 21, 24, 22, 21 and 20 respectively (Figure 3C). During the whole cold storage period, esters were the most abundant, followed by aldehydes, and were the least ketones, (Figure 2B) and contributed the most to the aroma of apple during cold storage, mainly hexyl esters and acetate esters. Some of the main compounds were hexyl 2-methylbutyrate, hexyl butyrate, hexyl hexanoate, butyl butyrate, 2-hexenal, hexanal, and 1-Hexanol (Appendix A).

### 3.4. Bioinformatics and Phylogenetic Analysis of MdCXE Gene Family Proteins

We identified 42 members of the apple CXE family from the GDR based on the published genome of ‘Golden Delicious’ apple, and found that all 42 MdCXE CDS contain complete predicted coding regions. Furthermore, the amino acid sequences of these proteins exhibit similar characteristics to those of the previously reported 20 AtCXE gene family members in *A. thaliana*. (*AtCXE1-20*, [19]), including the transmembrane region of the α/β hydrolase fold in the transmembrane region of the superfamily. As well as , oxygen anion pore and catalytic triad residues (Ser, acid, and His) predicted with conserved HGG sequence motifs, including the pentapeptide sequence around the nucleophilic serine (GXSXG) (Appendix A)

According to the location of the selected genes on the chromosome, we systematically named them as *MdCXE1*-*MdCXE42* (Appendix A). We performed phylogenetic analysis on the predicted 42 apple CXE genes, 4 tomato CXE genes with known functions, 3 peach CXE genes and 20 *AtCXEs* genes previously reported (Figure 4). According to the classification of Arabidopsis [19], 42 members were divided into 7 clades (Group 1–7). In addition, as revealed by Blastp search, MdCXE has close similarities with carboxylesterases in other higher plants, ranging from 50% to 100%.

### 3.5. MdCXE Gene Expression Analysis

To identify additional genes associated with carboxylesterase hydrolysis, we conducted a blast alignment analysis using CXE genes from various plants and identified 10 genes with 100% homology to them. The genes involved in *MdCXE3*, *MdCXE5*, *MdCXE6*, *MdCXE9*, *MdCXE10*, *MdCXE12*, *MdCXE17*, *MdCXE20*, *MdCXE23*, and *MdCXE25*. During cold storage, the expression of these 10 candidate genes was analyzed by RT-qPCR using ‘Ruixue’ fruit as material, and a family member with high expression was selected as a candidate gene.

The results showed that *MdCXE10* was not expressed during refrigeration. However, *MdCXE20* had a significant high transcription level during fruit cold storage (Figure 5B); In addition, the phylogenetic tree (Figure 4) shows that *MdCXE20*, like other plant carboxylesterases, belongs to the Group3 group and is the largest branch. Previous studies have shown that the Group3 group contains the volatile ester *SlCXE1* involved in tomato and the volatile ester *PpCXE1* involved in peach fruit (Figure 4). Based on the above analysis, it is speculated that *MdCXE20* may be involved in the degradation of apple volatile esters.

Esters account for the highest proportion in ‘Ruixue’ apple varieties (Figure 3B). In order to further explore the degradation of volatile esters during storage, the contents of main characteristic esters, butyl butyrate, hexyl hexanoate, hexyl 2-methylbutyrate, and hexyl butyrate, C4 and C6 alcohols as precursors of ester synthesis and the expression of carboxylesterase *MdCXE20* gene involved in ester degradation were further analyzed. The results showed that the peak of hexyl caproate appeared at 120 d, and the peak of the corresponding substrate 1-Hexanol content was delayed. Similarly, the content of butanol peaked at 90 d, and the corresponding butyl ester peaked at 60 d. However, there was no significant correlation between hexyl 2-methylbutyrate, hexyl butyrate content and hexanol content. It was observed that the expression of *MdCXE20* gene in fruit decreased with the increase of esters, and it was speculated that the carboxylesterase *MdCXE20* may degrade some hexyl ester and butyl ester.

In order to explore the correlation between the candidate gene *MdCXE20* and the content of volatile esters, the correlation between the expression level and the content of volatile esters in ‘Ruixue’ fruit during cold storage was analyzed by RT-qPCR. Correlation analysis revealed that the transcription level of *MdCXE20* was negatively correlated with the content of esters (Figure 3C), and the correlation coefficient R^2^ of volatile ester content of *MdCXE20* gene expression was 0.9389 (*p* < 0.05).

### 3.6. Transient Overexpression and Silencing of MdCXE20 Changed the Content of Esters in Apple Fruit

Homologous transient overexpression was used [35] and virus-induced gene silencing (VIGS) system to verify the function of *MdCXE20* in volatile substances in apple fruits. In this study, we created pGreen-OE-*MdCXE20* and TRV1 + TRV2-*MdCXE20* vectors, with pGreen-OE and TRV1 + TRV2 vectors as controls. These vectors were then injected into mature apple fruits for further analysis (Figure 6A). The expression of *MdCXE20* gene and the changes of VOC were detected 7 days after injection. As shown in Figure 6B, after overexpression of *MdCXE20* gene, the expression level increased significantly by about 2.5 times, the corresponding total volatile substances decreased by 2 times, and the esters decreased by about 1.5 times, while silencing *MdCXE20*, the gene expression decreased significantly. It decreased by about 0.8 times, and the esters increased by about 1.5 times. The contents of butyl butyrate, hexyl hexanoate, hexyl 2-methylbutyrate and hexyl butyrate in apple fruits of *MdCXE20* overexpression decreased significantly, and the opposite results were observed after silencing *MdCXE20* (Figure 6C). The above results indicated that *MdCXE20* gene was involved in the degradation of esters in apple fruit.

### 3.7. MdCXE20 Transgenic Reduced the Content of Ester Volatile Substances in ‘Wanglin’ Callus

In order to further confirm the function of *MdCXE20* on ester volatiles, we performed homologous stable transformation experiments in ‘Wanglin’ callus. Agrobacterium containing pGreen-OE-*MdCXE20* construct was used to infect ‘Wanglin’ callus, and wild ‘Wanglin’ callus was used as control (Figure 7A). Three independent lines, *MdCXE20*-#OE1, *MdCXE20*-#OE2 and *MdCXE20*-#OE3, were obtained. Subsequently, the DNA detection and Western Blot detection of the three transgenic lines could detect the target band, while the control could not detect the target band (Figure 7C,D). The gene expression level in the three transgenic lines of overexpression *MdCXE20* was significantly higher than that in the wild-type ‘Wanglin’ callus (Figure 7B), the most significant was the *MdCXE20*-#OE3 line, the transcription level was about 18 times that of the control group, and SPEA-GC-MS analysis was performed. After overexpression of *MdCXE20*, the content of esters decreased by 78% (Figure 7E). In summary, these results indicate that overexpression of *MdCXE20* is involved in the degradation of ester volatiles in plants.

### 3.8. Subcellular Localization Analysis

In order to determine the subcellular location of *MdCXE20* in apple, we verified the hypothesis that *MdCXE20* protein is located on the plasma membrane of plants by the expression of *MdCXE20* and GFP fusion protein. We transiently transformed Agrobacterium tumefaciens carrying pCAMBIA2300-GFP-*MdCXE20* and control construct pCAMBIA2300-GFP into tobacco leaf epidermal cells containing transgenic Mcherry. After injection, the leaf slices around the injection site were observed by laser scanning confocal microscopy after dark culture for 3 days. As shown in Figure 8, the control pCAMBIA2300-GFP constructs were distributed throughout the cell, while the GFP-*MdCXE20* protein fluorescence was targeted to the plasma membrane. These results indicate that *MdCXE20* is located in the cytoplasm where the hydrolysis of aromatic esters is likely to ocuur, thereby exerting its biological function.

## 4. Discussion

In China, apple is widely recognized as a key economic fruit tree. It serves as a major source of income for fruit farmers and has significantly contributed to the promotion of rural revitalization [38]. After harvesting, fruits are susceptible to rot and quality decline due to high water content and inadequate storage measures. This can result in significant economic losses. To address this issue and ensure year-round supply, refrigeration has become the most popular fruit storage method worldwide. Cold storage is a widely used and simple method for preserving fruits and vegetables. By lowering the temperature, the respiration rate and ethylene release rate of fruits are inhibited, which helps to maintain fruit firmness and achieve optimal preservation [39]. ‘Ruixue’ apple is a late-maturing variety that exhibits greater resistance to storage. Our study found that fruit firmness and crispness exhibited a significant decrease during storage, which is consistent with previous research.

Texture and volatile aroma compounds are both important factors in determining fruit quality and affecting final consumer acceptance of goods. These traits are crucial in quality, disease resistance breeding [40], and post-harvest processing and storage [41]. Metabolic pathways for volatile aroma substances are extensive and intricate, with numerous precursors and enzymes involved in the process. In plants, fatty acids and amino acids are typically the metabolic precursors. Key enzymes responsible for aroma metabolism include LOX, HPL, PDC, ADH, and CXE, among others. Studies have revealed a significant increase in hexyl esters and butyl esters in apple volatiles after treatment with hexanal and hexanoic acid [42]. The synthesis of aroma in melon fruit was significantly impacted by the cloning of *CmADH1* and *CmADH2*, as reported in [43]; Similarly, the production of VOC in ‘Pink Lady’ apples was stimulated by treatment with n-butanol during the early stages of development [44]; Exogenous fatty acid treatment of apple fruits resulted in a significant increase in hexyl esters and butyl esters, indicating that fatty acids are precursors of aroma synthesis. Additionally, the activity of LOX was found to be related to aroma synthesis, as it increased in response to the fatty acid treatment [42]. Although the activity of alcohol dehydrogenase involved in ethanol formation was not found to be related to ethylene regulation [45], a study showed that the content of total and monools in apples was induced by ethylene and inhibited by ethylene receptor inhibitor 1-MCP [10]. This suggests that volatile aroma compounds in apples are typically associated with substrate utilization and related enzyme activity [42]. This study examined the impact of aroma synthesis-related enzymes, including LOX, HPL, ADH, and PDC, on the quality of stored produce. Results indicated that the LOX-HPL enzyme activity remained consistent, with a significant increase observed before 90 days of cold storage. The production of volatile esters showed a gradual increase, peaking at 60 days, which was 30 days later than the peak in enzyme activity. It is speculated that it may be related to substrate content and availability of other compounds, and further research is needed.

The structure of volatile aroma substances in apples is complex and their quantity and distribution are influenced by various factors such as varieties, environment, cultivation, postharvest storage, and other conditions [46], particularly postharvest factors. Among the VOCs, only a few have a significant impact on fruit aroma, mainly esters, aldehydes, alcohols, and terpenes. Esters are considered the most important volatile substances in apple aroma [47,48]. During the cold storage of ‘Ruixue’ apple fruit, the content and types of volatile aroma substances underwent a dynamic change. As storage time increased, the aroma substances showed a trend of initially increasing and then decreasing, with the peak being reached at 60 days of storage. The esters constituted the largest proportion of the volatile substances, accounting for approximately 50% of the total. The study detected several main esters, including butyl butyrate, hexyl acetate, 2-methylbutyrate, hexyl hexanoate, hexyl thiocyanate, and isoamyl 2-methylbutyrate. The dominant esters were found to be hexyl esters, which is in line with previous research [4]. In the ‘Ruixue’ variety, compounds such as ethyl butyrate, butyl propionate, and propyl hexanoate were not detected until after 30 days of storage, indicating significant variations in the content and types of aromatic substances during different storage periods following cold storage.

Esters are significant volatile aroma compounds found in many mature fruits and are derived from amino acids and fatty acids. LOX-HPL catalyzes fatty acids to produce C6 and C9 aldehydes, which are then converted to alcohols through the dehydrogenation of ADH [42,49,50]. Finally, esters are produced through the action of AAT [51]. Tomato *SlAAT1* and peach *PpAAT1* are examples of enzymes related to the production of volatile esters during fruit ripening [52]. The synthesis of esters has been extensively studied in relation to the quality of apple fruit. However, there is a lack of research on the metabolic processes of esters during storage. This area presents an opportunity for further investigation and could provide valuable insights into the factors that influence the quality of stored apples.

CXEs, as a member of the α/β hydrolase superfamily, are known to catalyze various hydrolysis reactions and are typical ester bond hydrolases. However, there have been few studies on the hydrolysis of ester volatiles in fruits by carboxylesterase. The functions of some CXEs in plants, such as tomato, peach, and strawberry, have been identified [26]. The role of CXE protein in regulating the content of volatile esters in tomato has been widely studied, where *SlCXE1* has been found to hydrolyze volatile esters [26]; *PpCXE1* is known to be involved in the hydrolysis of volatile esters [25]. In this study, comparative analysis identified 42 members of the apple MdCXE gene family, and it was suggested that 10 of these genes may be responsible for the hydrolysis of volatile esters during the storage of ‘Ruixue’ fruit. The study confirmed that the gene *MdCXE20*, which exhibited the highest transcription level, plays a crucial role in the hydrolysis of volatile esters. Through homologous transient and stable transformation of ‘Wanglin’ callus, it was established that *MdCXE20* is involved in the decomposition and metabolism of volatile esters that are related to the flavor of apple fruit. Similar studies on tomato and peach fruits have revealed that acyltransferases AATs and CXEs work together to regulate the content of volatile esters in fruits [25,26,52]. During fruit storage, the level of CXE transcriptional metabolites significantly increased. Subcellular localization analysis revealed that *MdCXE20* and *MdAAT1* gene were both located in the cytoplasm [53]. The co-localization of these enzymes can create a cycle that regulates the balance of ester synthesis and hydrolysis in fruits through synergistic action [52]. The study revealed that the *MdCXE20* gene is capable of hydrolyzing esters. However, due to the current limitations in gene silencing or knockout transgenes in apple fruits for functional verification, it is difficult to determine whether other CXE gene members also play a role in regulating ester content.

Our research has identified *MdCXE20* as a crucial gene in the synthesis pathway of ester volatiles in apple fruit. The significant expression of *MdCXE20* in apple fruit has been found to cause a decline in hexyl ester content. This alteration in volatile substances has a direct impact on the aroma of the fruit, ultimately affecting its overall sensory profile. In summary, this study offers a comprehensive understanding of the formation and regulation of fruit aroma quality. It achieves this by exploring the regulation mechanism of ester aromatic substances in apple fruit. The findings provide an important theoretical basis for improving fruit aroma quality through molecular breeding and other methods.

## 5. Conclusions

During cold storage, ‘Ruixue’ apples are mainly composed of hexyl esters such as hexyl acetate, hexyl caproate, and hexyl thiocyanate which are the most important volatile substances after harvest. Carboxylesterases (CXEs) is a diverse and complex enzyme group that is capable of hydrolyzing esters. However, their natural substrates are not well understood. In this study, 42 MdCXE family members in apples were identified through phylogenetic analysis. The results of RT-qPCR revealed that *MdCXE20* had a high expression level in apple. The results from the transient overexpression of *MdCXE20* in apple fruit in *E. coli* and the stable overexpression of ‘Wanglin’ callus indicate that *MdCXE20* functions in the degradation of esters. These findings suggest that *MdCXE20* plays a crucial role in ester metabolism in mature apples, providing valuable insights into the potential role of CXE family members in apples and enhancing our comprehension of the molecular basis of fruit volatiles.

## Figures and Tables

**Figure 1 foods-12-01977-f001:**
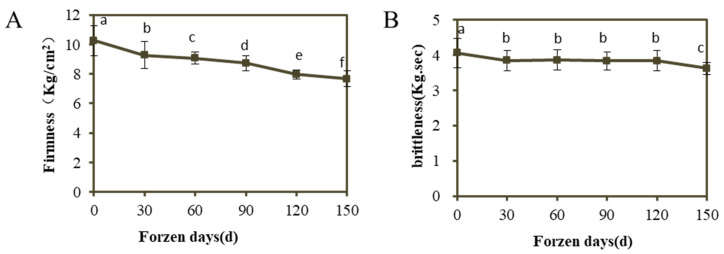
Changes of firmness and brittleness of apple fruit during storage. (**A**) Fruit firmness; (**B**) Fruit brittleness. Error bars show ± SE from three biological replicates. Different lowercase letters in columns denote significant differences between sampling dates for ‘RuiXue’ fruit by Duncan’s multiple range test (*p* < 0.05).

**Figure 2 foods-12-01977-f002:**
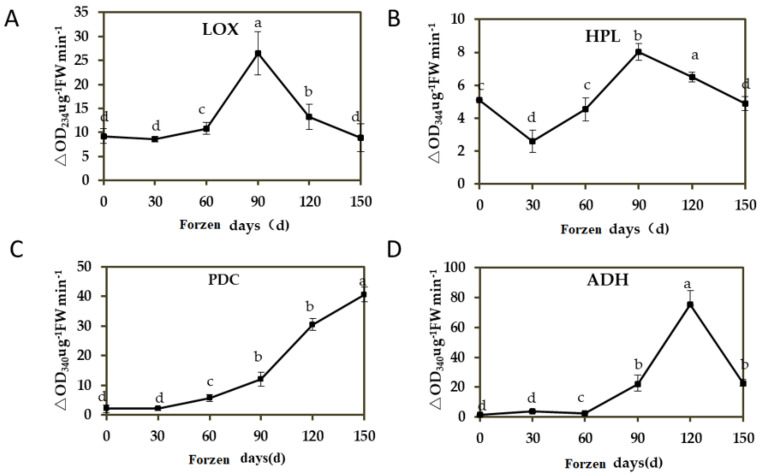
Changes of aroma synthase during apple cold storage. (**A**) Lipoxygenase (LOX); (**B**) Hydroperoxide lyase (HPL); (**C**) Keto acid decarboxylase (PDC); (**D**) Acetaldehyde dehydrogenase (ADH). Error bars show ± SE from three biological replicates. Different lowercase letters in columns denote significant differences between sampling dates for ‘RuiXue’ fruit by Duncan’s multiple range test (*p* < 0.05).

**Figure 3 foods-12-01977-f003:**
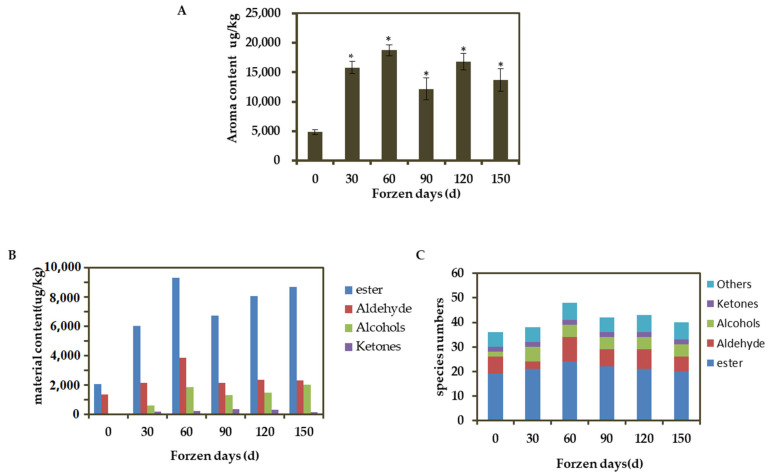
VOC in apple fruits during different cold storage periods. (**A**) Total volatile content, * indicates that the difference is statistically significant at the *p* < 0.05 level. Error bars were calculated from three biological experiments, and show the standard deviation of the mean; (**B**) The content of VOC in each component; (**C**) Number of VOC. Different colors represent different types of volatile substances.

**Figure 4 foods-12-01977-f004:**
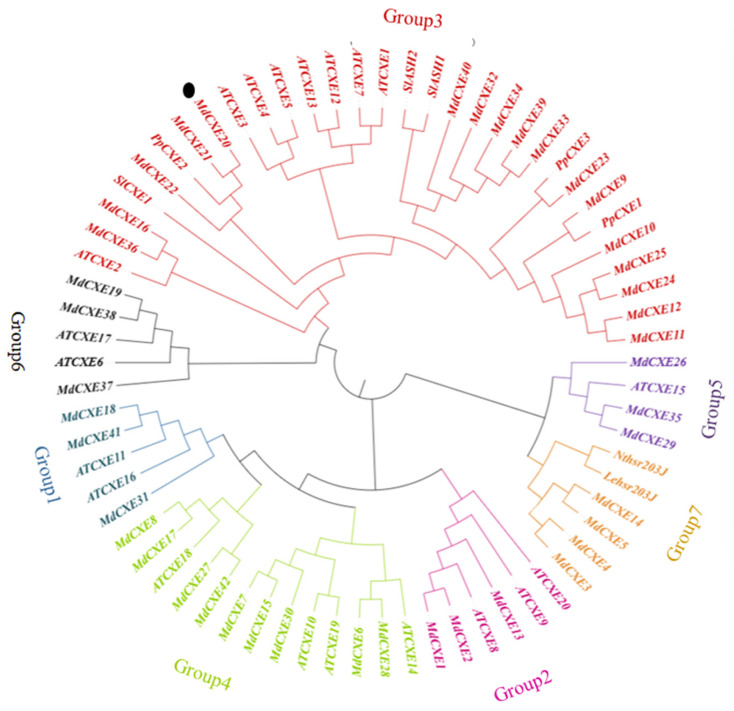
Phylogenetic analysis of CXE families. The full-length amino acid sequences were aligned to construct the phylogenetic tree. The accession members are *AtCXE1-20* from *Arabidopsis thaliana*, *SlCXE1*, *SlASH1-2* and *Lehsr203J* from tomato, *PpCXE1-3* from peach fruit and *Nthsr203J* from tobacco. Different colors represent different groups.

**Figure 5 foods-12-01977-f005:**
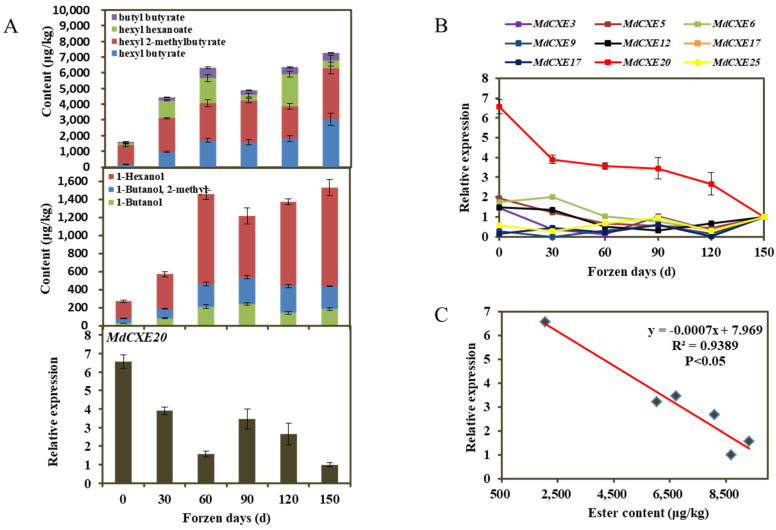
MdCXE gene expression analysis. (**A**) The content of volatile esters and alcohols and the expression of *MdCXE20* gene during apple cold storage. Error bars represent SE (*n* = 3); (**B**) Expression pattern of apple CXE gene during cold storage; (**C**) Transcript levels of CXEs during fruit cold storage which showed significant negative correlation to volatile esters (*p* < 0.05).

**Figure 6 foods-12-01977-f006:**
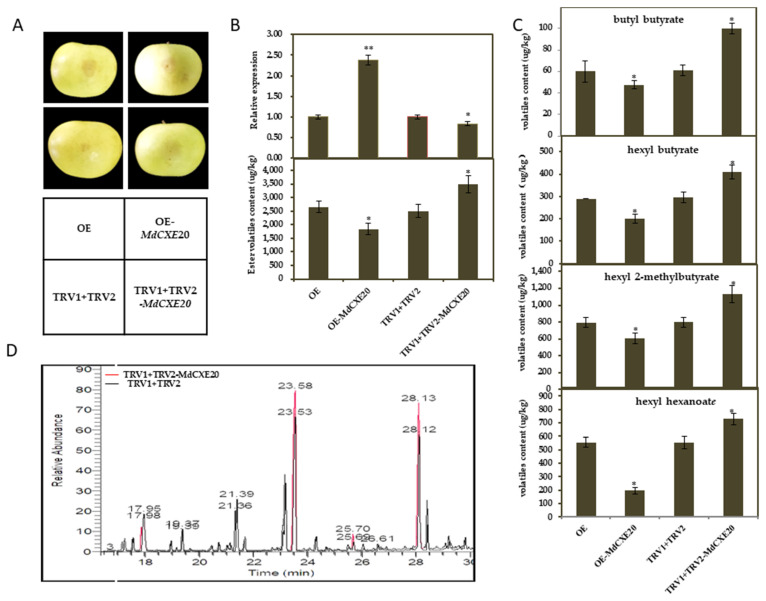
Instantaneous injection of *MdCXE20* ‘Ruixue’ fruit. (**A**) ‘Ruixue’ apple transient injection; (**B**) ‘Ruixue’ fruit *MdCXE20* gene expression and ester content change; (**C**) Instantaneous expression of *MdCXE20* ‘Ruixue’ apple 4 ester content; (**D**) Chiral GC-MS analysis of four esters in transiently silenced apple fruit, Red was virus-induced gene silencing, and black was blank control. Single and double asterisks (*), (**) represent significant differences (* *t*-test, *p* < 0.05; ** *t*-test, *p* < 0.01).

**Figure 7 foods-12-01977-f007:**
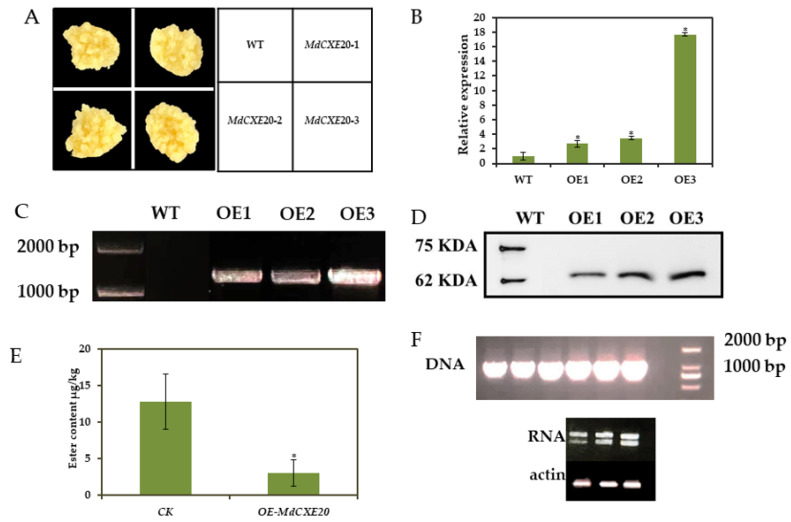
Changes in transcription level and ester content of *MdCXE20* transgenic ‘Wang Lin’ callus. (**A**) Schematic diagram of ‘Wang Lin’ callus; (**B**) Transcription level of *MdCXE20* transgenic ‘Wang Lin’ callus; (**C**,**D**,**F**) RNA, DNA, protein level identification of transgenic materials; (**E**) Change of ester content of *MdCXE20*-#OE3 transgenic ‘Wang Lin’ callus. WT: wild-type; *MdCXE20*-#OE1, *MdCXE20*-#OE2, *MdCXE20*-#OE3 a transgenic lines. Double asterisks (*) represent significant differences between wild-type and transgenic callus (*t*-test, *p* < 0.01).

**Figure 8 foods-12-01977-f008:**
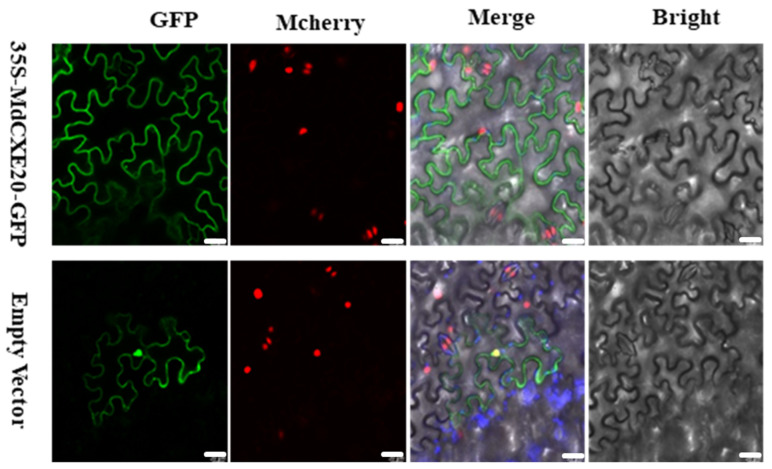
Subcellular localization of *MdCXE20*. GFP signals were observed in *Nicotiana benthamiana* plants. The pCAMBIA2300-GFP empty vector was used as a control. Scale bar = 10 µm.

**Table 1 foods-12-01977-t001:** Lists of all the primer sequences used for RT-qPCR.

Name	Apple ID	Primers (F)	Primer (R)
*MdCXE3*	MD02G1275900	TGATGGCTCTGTCGACCGG	AAATATATTCGGACACGATGG
*MdCXE5*	MD02G1276600	TGGATTCGTTCGATCAGT	TAAAGACGGTGCTCTGGA
*MdCXE6*	MD03G1273300	CTCGTTTGATCACGTCGACAC	CTGCTCAAGATCCGAAATGCC
*MdCXE9*	MD05G1076400	AAGAACCAGCATTGTCCGT	AATGCAGTTTCAACCACGAA
*MdCXE10*	MD05G1078900	TGCCTACAATCTCCATACTCG	ATGAAACGCTCAATTGTACCA
*MdCXE12*	MD05G1191100	CAACCCGAAACCGGAGTCC	GTGGGGGAGGAAGCGCTTTC
*MdCXE17*	MD08G1226300	ATCATTCGAGTTCCCGACCCT	CTTTGGATTGGACCCCGGTT
*MdCXE20*	MD10G1068500	GAGGAGGTACTACCAGTGGTTGAAG	TCACACGGAAGCAATGAAATCTTTG
*MdCXE23*	MD10G1091200	ATGGATTCAGCCTTGAGCAAC	ACATTCGACGCCTGTTTGTG
*MdCXE25*	MD10G1091900	ATGACAAACGAAGTAGCCCAT	CCTTGGATTGGACACCGATT
*MdActin*	MD01G1001600	GATATCTCCACTGACGTAAGGGATG	AGGGTCAGCTTGCCGTAGGTGGCA

**Table 2 foods-12-01977-t002:** Construction of pCAMBIA2300 and TRV2 primers by *MdCXE20*.

Gene	Vector	Primers Sequences
*MdCXE20*	pCAMBIA2300-OE-Primers (F)	acgggggacgagctcggtaccATGACGACGTCGTTGGACTCC
	pCAMBIA2300-OE-Primer (R)	ggtgtcgactctagaggatccCACGGAAGCAATGAAATCTT
*MdCXE20*	TRV2-Primers (F)	gtgagtaaggttaccgaattcTTTTTCGGGGGAGAGGAGC
	TRV2-Primer (R)	gagacgcgtgagctcggtaccCACGGAAGCAATGAAATCTT

## Data Availability

The datasets generated for this study are available on request to the corresponding author.

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
