# Peer review of "Changes in the VOC of Fruits at Different Refrigeration Stages of ‘Ruixue’ and the Participation of Carboxylesterase MdCXE20 in the Catabolism of Volatile Esters"

_foods, 2023, doi:10.3390/foods12101977_

Round 1

Reviewer 1 Report

The current manuscript describes an interesting work aimed to understand in detail the involvement of carboxylesyterases (MdCEX20) in the metabolism of volatiles esters that are a main component of the aromatic fraction of apple fruit. Authors carried out an exhaustive molecular study, identifying a number of MCEX20 gene from the apple genome, analysed the expression of several gene members they also made transgenic apples overexpressing and suppressing MdCEX20 gene with the corresponding volatile analysis. In my opinion, this is an interesting and relevant work that still requires revision and editing, and improvement of the English style.

Following is a list of specific comments:

-Remove from the abstract details of introduction (line 11-14).

- Line 18. Expression higher?. Complete the sentence, compared to what?

- Line 36. Rephrase the sentence; the importance of the crop is not the related to the complex and number of volatiles.

- Used throughout the manuscript VOC (volatile organic compounds) that is a widely used and recognized initial for volatile compounds.

- line 40, replaced change by impact.

- line 67, define the function of the CXE enzyme activity.

- Line 72, indicate “few studies…” but relevant studies are mentioned below.

- line 83, Italics by Arabidopsis thaliana

- Line 96, apple variety

- Line 99-105., please define clearly the objective of the study, what was the purpose of the study, and then what it has been done.

- Provide detail of the refrigeration temperature, time of sampling. 

- line 126, I think the term Firmness is more adequate than hardness.

- Provide a better description of table 2 legend.

- Line 244, Instantaneous transformation, is that a correct sentence???

- Line 245. Method reference?

-Line 306: What is brittleness? It has not been described in material and methods.

- Are data of F1g 1A and B statistically significant? I suggest showing the data in a table with significance analysis.

- Describe first data Fig 1A and then Tig 1B.

- Line 329-324, where these data are reported?

- Line 332, FIg. 2, provide full description of the initials.

- Line 342. …esters during cold storage…

- Line 357, I do not think are cDNA, sequences.

- Line 359. Rephrase the sentence.

-Lone 367, 42, not 44, apple CXE genes

- Line 368, Fig 4 not 3.

- Line 378, is that sentence correct, rephrase

- Line 380, The genes involved in…

- Line 387, description of Fig. 4 before 5

- I think that the paragraph starting in line 393, shul be in new section (probably 3.6), separating gene expression results from that of volatile emission.

- Fig. 5A. axis of the figure are “forzen days”. I suggest to uniform with Fig 2 “storage days”

- Line 418, remove first sentence is not correct as the are many reports of transgenic apple (not fir that specific gene, but many others).

-Line 425-428, rephrase “….overexpression ….2.5–times and the …. while siliencing…..”.

- Line 435. Instantaneous injection???

- Line 162 Subcellular

- line 469, Fig 8, not 3-4

- -line 472, located in the plasma membrane where the hydrolysis of….is likely to ocuur.

- Line 480. Remove first sentence of the discussion, irrelevant, not in recent years not only in climacteric fruits…

- Line 487-489, rephrase the sentence.

- Line 489. The current study…

- Line 521. Provide references from this statement.

- Line 539, not Cytosolic, in previous section they indicate MdCXE20 is located in the plasma membrane.

- Line 547. I do not think “key regulator” in the correct term, key gene in the pathway…

- Discussion should be improved, they do not discuss about the structure of the genes, sequence similarities with the same gene of other plants. Moreover, the potential involvement in VOC emission during storage should be also discussed, a in general, discussion shild be substantially improved.

Author Response

Dear reviewer:

    Wish you a happy life.

Reviewer 2 Report

Dear Authors,

The manuscript being presented possesses both practical and scientific value. However, at its current state it is rather confusing for the reader.

Below you can find several suggestions that may help you organize it better:

- check for grammatical and lexical errors, as well as avoid repetition like in L30-31 for example.

- the Introduction is unnecessary long, some of the information can be reorganized in the Discussion section.

- the results about the hardness seem neglected compared to other results.

- esters are much studied, so I honestly believe that you can highlight some of the key components that are specific for the cultivar you have studied.

- you references seem as if plotted without software, please check the instructions for authors and add the necessary information in this Section.

- the conclusion section could be enriched (for example you have placed much more information in your Abstract, than in your conclusion).

- do you an opportunity for future studies based on the current results?

Author Response

Dear reviewer:

    Wish you a happy life.

Round 2

Reviewer 1 Report

This manuscript is a revised version of a previos submission that I also reviewed. In my opinion, authors have addressed most of the comments and sugestions and the manuscript is now subtantialy improved. My only concern is that a revission of the english style and additional tunning, would improved the clearlity of the manuscripts.

Author Response

Dear reviewer:

    We are very grateful to your comments and suggestions for the manuscript. These comments are of great help for improving our paper.According to your valuable advice, we modified the English style and English format in the original manuscript. Partially modified as follows:

  1. We processed all the gene names and species Latin names with italics.
  2. The specific components of ester aroma vary depending on the type of fruit. Apples are mainly composed of two types of esters, namely, branched chain esters like 2-methylhexyl butyrate and 2-methylbutyl acetate, and linear esters such as hexyl butyrate and hexyl acetate [10, 11](Line 57-60).
  3. The determination was performed using Yang's modified SPME-GC-MS method [31] (line 190).
  4. After RNA extraction, cDNA was synthesized using the EasyScript One-Step gDNA Removal and cDNA Synthesis Kit according to the manufacturer's instructions. The expression level of the candidate gene carboxylesterase, which is involved in the degradation of aroma esters, was analyzed using fluorescent quantitative real-time PCR (RT-qPCR) with cDNA as the template (line 221-225).
  5. In this study, we created pGreen-OE-MdCXE20 and TRV1+TRV2-MdCXE20 vectors, with pGreen-OE and TRV1 + TRV2 vectors as controls. These vectors were then injected into mature apple fruits for further analysis (Figure 6A) (line 427-429).

  We are grateful for the reviewer’s comments and suggestions to improve our manuscript greatly. We hope that the corrections will meet with approved. Thank you.

   Wish you a happy life!

Dongmei Li

2 May 2023 

Reviewer 2 Report

The authors have provided answers to the comments from the 1st round of revision. The manuscript is somehow revised.

Author Response

(The authors gave the same response as above.)
